# Inhibition of TRPV1 channels by a naturally occurring omega-9 fatty acid reduces pain and itch

Sara L. Morales-Lázaro[1], Itzel Llorente[1], Félix Sierra-Ramírez[1], Ana E. López-Romero[1], Miguel Ortíz-Rentería[1], Barbara Serrano-Flores[1], Sidney A. Simon[2], León D. Islas[3] & Tamara Rosenbaum[1]

The transient receptor potential vanilloid 1 (TRPV1) ion channel is mainly found in primary nociceptive afferents whose activity has been linked to pathophysiological conditions including pain, itch and inflammation. Consequently, it is important to identify naturally occurring antagonists of this channel. Here we show that a naturally occurring mono-unsaturated fatty acid, oleic acid, inhibits TRPV1 activity, and also pain and itch responses in mice by interacting with the vanilloid (capsaicin)-binding pocket and promoting the stabilization of a closed state conformation. Moreover, we report an itch-inducing molecule, cyclic phosphatidic acid, that activates TRPV1 and whose pruritic activity, as well as that of histamine, occurs through the activation of this ion channel. These findings provide insights into the molecular basis of oleic acid inhibition of TRPV1 and also into a way of reducing the pathophysiological effects resulting from its activation.

[1] Instituto de Fisiología Celular, Universidad Nacional Autónoma de México, Circuito exterior s/n, Coyoacan 04510, Mexico. [2] Department of Neurobiology, Duke University, 327C Bryan Research Building, Durham, North Carolina 27710, USA. [3] Departamento de Fisiología, Facultad de Medicina, Universidad Nacional Autónoma de México, Circuito escolar s/n, Coyoacan 04510, Mexico. Correspondence and requests for materials should be addressed to T.R. (email: trosenba@ifc.unam.mx).

The two primary strategies targeted to control and treat pain have concentrated on preventing the propagation of action potentials in peripheral nociceptors from reaching the central nervous system, and identifying and then inhibiting the receptors whose activation will result in the generation of said action potentials. One such protein is the transient receptor potential vanilloid 1 (TRPV1), whose role in inflammatory and neuropathic states is well established[1].

TRPV1 is a non-selective cation[2] channel that is activated by diverse stimuli including capsaicin, noxious temperatures (near 42 °C), extracellular acidic pH[3] and bioactive lipids such as lysophosphatidic acid (LPA)[4], all of which have been shown to activate nociceptors. In investigating the structure–activity profile of LPA on TRPV1, we showed that lysophospholipids that activated the channel exhibited specific structural requirements regarding their head group and acyl chain composition[5]. In this process, we found that oleic acid (OA; cis-9-octadecenoic acid), a naturally occurring long-chain monounsaturated fatty acid, which resembles LPA in both acyl chain composition and being negatively charged, does not activate TRPV1 but rather acts as a naturally occurring inhibitor of this channel. Although many synthetic antagonists have been developed, only a few natural inhibitors have been identified[6–8]. Given TRPV1's role in pain and inflammation, a clinically relevant goal has been to identify natural antagonists of the TRPV1 channel. Moreover, it is important to understand how similar molecules could lead to either the activation or inhibition of this channel and thus affect physiological processes such as pain, inflammation and itch[3,9].

Here we show that the inhibition of TRPV1 by OA through a direct interaction with the vanilloid-binding pocket (VBP) also results in the inhibition of TRPV1-dependent pain and itch, and that inhibition arises from the stabilization of a closed state of the channel and thereby precludes its activation in response to several stimuli. Moreover, we describe a novel endogenously present itch response inducer, cyclic phosphatidic acid (cPA), which activates TRPV1 and show that this physiological response, as well as itch produced by histamine, are also inhibited by OA. These findings provide insights not only into the molecular basis of OA inhibition of TRPV1, but a novel way of reducing the pathophysiological effects of pain and itch resulting from its activation.

## Results

**OA inhibits heterologously expressed TRPV1 channels**. In HEK293 cells transfected with rat TRPV1 (rTRPV1), it was found that the application of a saturating capsaicin (Caps) concentration (4 μM) to inside-out membrane patches evoked the activation of currents that was reversible upon washing with recording solution (Caps and wash; Fig. 1a). Incubation of patches with 5 μM OA together with 4 μM capsaicin caused an $85 \pm 2.6\%$ reduction of the currents as compared with responses to 4 μM capsaicin alone (Caps + OA; Fig. 1a,b). The OA inhibition was largely reversible ($85 \pm 6.3\%$) on washing with capsaicin (Caps (reversibility); Fig. 1a,b). Moreover, the reversibility of OA inhibition progressed exponentially with a time constant of $38.8 \pm 6.5$ s. Finally, we note that under these conditions 5 μM OA did not activate TRPV1 at either depolarizing or hyperpolarizing voltages (Supplementary Fig. 1a,b) and that repeated applications of capsaicin (insets in Supplementary Fig. 1a,b) produced no desensitization of TRPV1 currents, under our experimental conditions.

To determine whether OA binds better to an open or closed channel state, it was applied in the presence or the absence of capsaicin. Thus, 5 μM OA was applied to inside-out patches expressing rTRPV1 either in an open state (at 60 mV in the

presence of 4 μM capsaicin applied right after capsaicin alone) or to a closed state (at $-60$ mV in the absence of any ligand). In both cases, near complete inhibition of capsaicin-activated currents was observed, although OA's inhibition was faster in the closed state ($\tau = 2.9$ s, Fig. 1c, blue circles) than in the open state ($\tau = 7.7 \pm 0.64$ s; Fig. 1c, black trace). Finally, in experiments with inside-out patches, we found that at 120 mV, OA inhibition after a 5 min incubation is concentration-dependent. The inhibition was quantified by fitting to the Hill equation with an apparent dissociation constant ($K_D$) of 2.2 μM and a Hill coefficient of 2 (Fig. 1d).

In another set of experiments, we found that capsaicin-activated currents were inhibited by 5 μM OA when it was applied either intracellularly (inside-out patches) or extracellularly (outside-out patches). Specifically, in inside-out patches, 5 μM OA inhibited the currents by $86 \pm 7\%$ and in outside-out patches by $77 \pm 6$ % (Fig. 2a–c; Supplementary Fig. 2a,b), with no statistical differences between both groups.

To determine whether OA could also inhibit the response to an endogenous TRPV1 activator we tested, in TRPV1-transfected HEK293 cells, its actions on the responses to LPA. In inside-out patches, 5 μM OA inhibited the LPA (5 μM)-activated current by $95 \pm 2\%$ (Fig. 2d,g). Another endogenous activator of TRPV1 is extracellular acidic pH, which acts at the extracellular surface[3,10]. In outside-out patches, we found that 5 μM OA inhibits pH 5.5 responses by $98 \pm 3\%$ (Fig. 2e,g).

We also tested whether 5 μM OA could inhibit the response to 5 μM cPA, a bioactive phospholipid that activates TRPV1 (ref. 5). We found that 5 μM OA inhibited cPA-activated currents by $79 \pm 5\%$ (Fig. 2f,g). For completeness, we also tested whether OA could inhibit the thermal activation of TRPV1 (ref. 11). Figure 2h–j shows that incubation of the TRPV1-expressing membrane patches with 5 μM OA decreased thermally activated TRPV1 currents by $72 \pm 8\%$ (Fig. 2i,j).

To ascertain whether fatty acids other than OA (C18:1 cis 9 omega-9) could inhibit TRPV1, we tested a variety of fatty acids that differed from OA in chain length (C16–C24), degree of unsaturation (0–3), location of double bonds, presence of a glycerol head group and cis or trans orientation of the C9 double bond (Supplementary Fig. 3a). Whereas 5 μM OA produced a marked inhibition of TRPV1 currents (Supplementary Fig. 3b), with the exception of another two naturally occurring compounds found in vegetable oils[12,13], petroselinic acid (C18:1 cis 6, 84% of inhibition) and linoleic acid (LA; C18:2 cis 9,12; 46% of inhibition), none of the other tested fatty acids had an inhibitory effect on TRPV1. These data, together with the fact that the channel was not inhibited by palmitoleic acid (PA; C16:1 cis 9 omega 7), a very similar molecule, demonstrate a stringent channel selectivity for OA.

**OA shifts the voltage dependence of TRPV1 currents**. We previously noted that OA inhibits TRPV1 activation in response to diverse stimuli, and that this inhibition occurs faster when OA is applied to channels in the closed state, indicating higher affinity for this configuration (Fig. 1c). For this reason, by measuring its effect on voltage and capsaicin activation of TRPV1, we tested whether OA would induce an allosteric effect on channel activation. We found that in the presence of capsaicin and after 40 s of OA application, the voltage activation of TRPV1 is markedly shifted to positive potentials by 108 mV, consistent with an allosteric effect of OA on the activation pathway (Fig. 3a–d). As the estimated charge ($q_a$) remains relatively constant around a value of 0.5 (Fig. 3c,d), we can calculate that the closed state is favoured by $\sim 1.2$ kcal mol$^{-1}$.

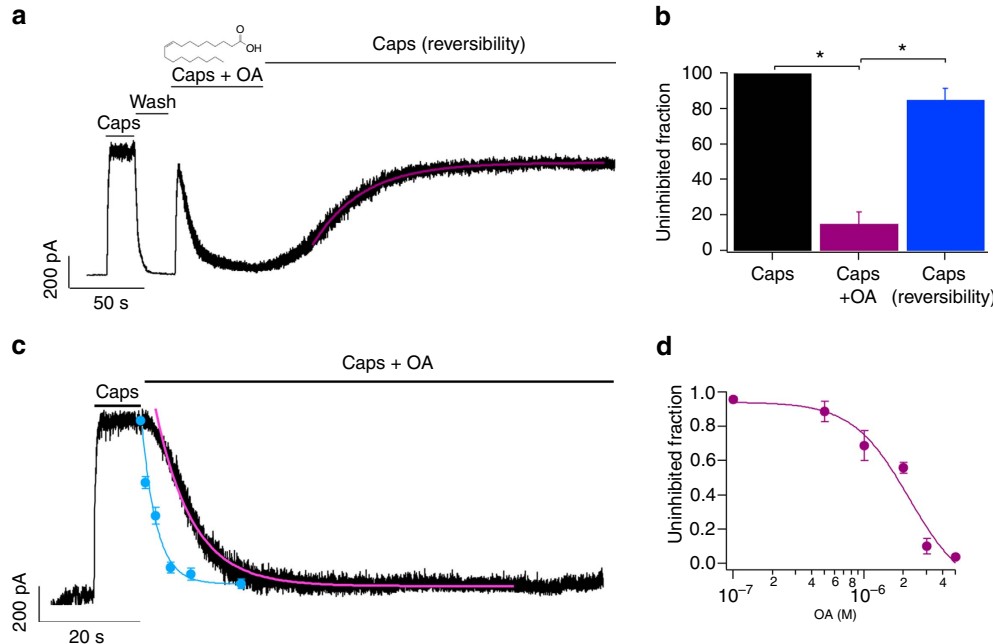

**Figure 1 | Heterologously expressed rTRPV1 channels respond to oleic acid. (a)** Representative trace ($n = 6$) of currents evoked at $+ 40$ mV for 350 ms from a holding potential of 0 mV. Inside-out patches were first exposed to recording solution, then to 4 µM capsaicin (Caps), washed with recording solution (wash), then exposed to 5 µM OA + Caps until inhibition was achieved. Capsaicin was then reapplied to measure recovery from inhibition (Caps (reversibility)). Data for recovery from inhibition were fit to a single exponential ($\tau = 38.8 \pm 6.5$ s). **(b)** Average data for experiment in **a**. Data were normalized to the initial value with capsaicin (black bar). *Denotes $P < 0.001$ between groups as compared by the brackets. After Caps + OA (magenta) and after Caps ((reversibility), blue), $15 \pm 2.6\%$ and $85 \pm 6.3\%$ of the currents remained, respectively ($n = 6$). **(c)** Time course of inhibition by 5 µM OA ($n = 5$) for the open (black) and closed states (blue). Data were obtained at 60 and $- 60$ mV and fit to a single exponential ($\tau = 2.9$ s and $\tau = 7.7 \pm 0.64$ s for the closed and open states; blue and pink lines, respectively). **(d)** Dose–response for inhibition of currents activated by 4 µM Caps by OA for 5 min (120 mV). Smooth curve is a fit with the Hill equation ($K_D = 2.2$ µM and Hill coefficient ($n$) $= 2$). Due to seal instability, a single OA concentration in the absence of capsaicin was tested per membrane-patch and the remaining capsaicin-activated current was normalized to the current obtained with capsaicin initially ($n = 5$ for each concentration point).

**OA stabilizes the closed state of TRPV1.** We next directly addressed the mechanism of OA inhibition of TRPV1 channels. Single-channel recordings in inside-out membrane patches show that in the presence of 4 µM capsaicin the open probability, $P_o$, is $\sim 1$ (Fig. 3e,f). However, co-application of OA with capsaicin decreases $P_o$ with a time constant of 6.2 s (Fig. 3e,f), without significantly changing the unitary channel current (Fig. 3e; Supplementary Fig. 4a–c). Kinetic analysis of the same single-channel records shows that the decrease of Po is a consequence of OA increasing the frequency and duration of long closed events and shortening the duration of open events (Supplementary Fig. 4d,e; Supplementary Table 1). As noted, this stabilization of the closed state can also be deduced from the 108 mV rightward shift in the conductance–voltage curve caused by OA. These data are consistent with the interpretation that OA allosterically stabilizes one or more of the channel closed states[14,15].

**Interactions of OA with TRPV1.** In lipid bilayers, the $pK_a$ of OA is 7.5 (ref. 16) and, consequently, at physiological pH, some OA molecules will be protonated and some will be negatively charged. The protonated OA molecules should be membrane permeable and, in principle, should be effective when added to either side.

Most of the other structurally similar compounds that were tested (Supplementary Fig. 3) did not exhibit the inhibitory properties of OA on TRPV1 activation. For this reason, we hypothesized that OA would bind to a specific site on TRPV1 (as opposed to producing its effect by altering membrane properties)[17,18].

To test this idea, we first performed experiments with several TRPV1 channels with mutations at positively charged sites located at different regions of the channel including the cholesterol-binding site[7]. As shown in Supplementary Fig. 5, mutation of these residues did not produce changes in the degree of OA inhibition as compared with the rat wild type (WT)-TRPV1 (rTRPV1) channel and even to the human TRPV1 (hTRPV1), which is also inhibited by this fatty acid.

It is established that due to the differences in some amino acids in the vanilloid-binding pocket (VBP)[19–21], chicken TRPV1 (CkTRPV1)[19,20] displays a reduced response to capsaicin. Relevant residues for the binding of capsaicin in TRPV1 include T550 in rat (which corresponds to A558 in the CkTRPV1 sequence (arrow in Fig. 4a)). A clue to the location of the OA-binding site came from experiments in HEK293 cells where we expressed capsaicin-insensitive CkTRPV1 channels. These channels are activated by pH 5.5 (ref. 3; Fig. 4b, left; Fig. 4d; Ck) and are only inhibited $29 \pm 13\%$ by 5 µM OA (Fig. 4b, right; Fig. 4d). In contrast, under the same acidic conditions, rTRPV1 currents are inhibited $89 \pm 2\%$ (Fig. 4c,d). Exposure of these channels to a vehicle (0.14% ethanol-containing solution) for 5 min produced no decrease in currents (grey bars, Fig. 4d), showing that channel run-down *per se* in rTRPV1 cannot account for the decrease in currents observed after treatment with OA.

We next investigated whether OA binds to the VBP[19–21]. This was accomplished first by performing a qualitative overlay assay that compared how OA competes with capsaicin or LPA (which does not act at the capsaicin-binding site[4]). As shown in Fig. 4e,f, TRPV1 binding to LPA was unaffected by the pre-incubation with capsaicin, whereas OA's binding was reduced in the capsaicin preincubated channels. Specifically, OA's interaction

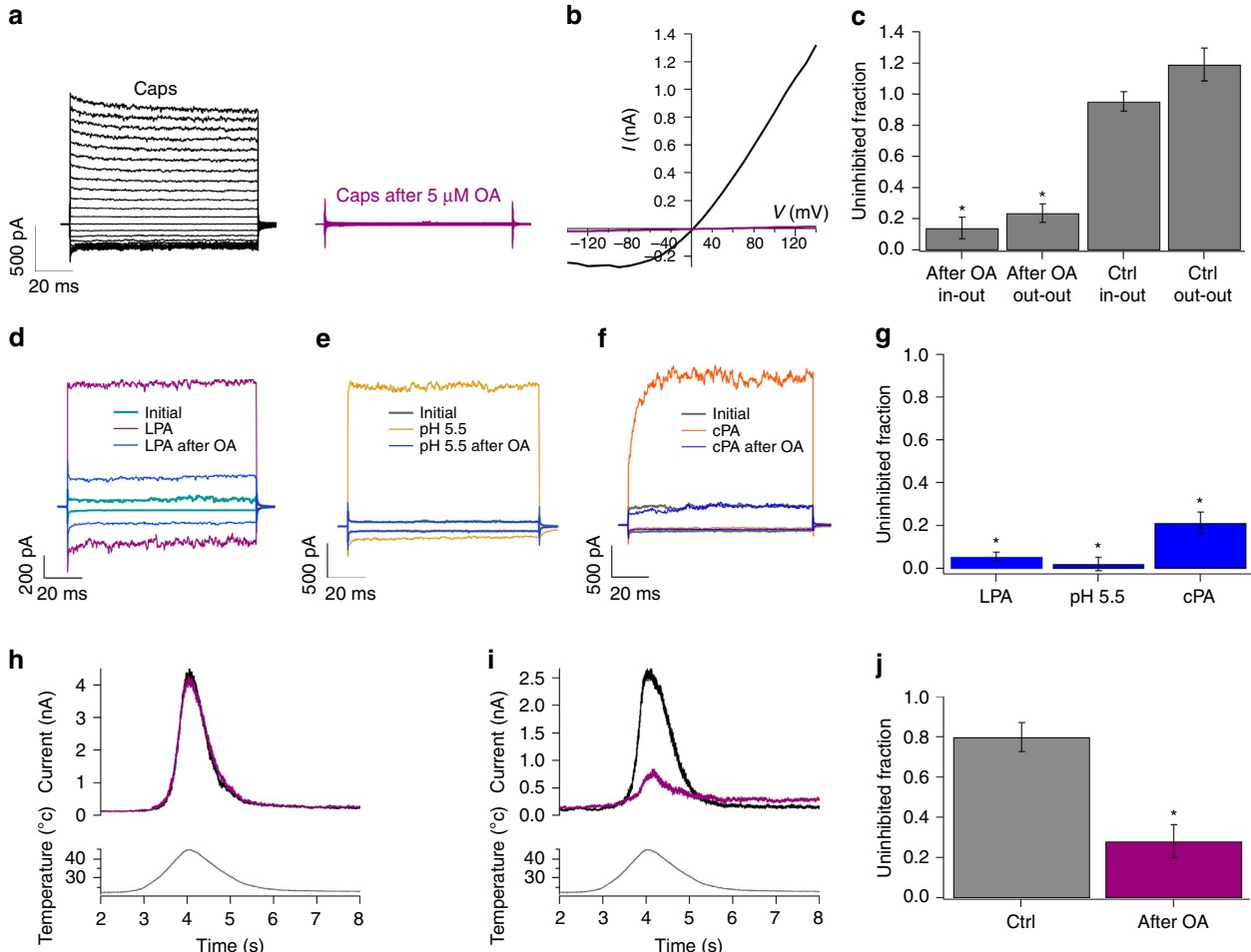

**Figure 2 | Oleic acid inhibits rTRPV1 currents activated by different stimuli.** (**a**) Inside-out patches were first exposed to 4 μM capsaicin (Caps), washed, and then to 5 μM OA for 5 min before re-exposing to capsaicin (magenta traces). (**b**) Current–voltage relationships for currents in **a** in the presence of 4 μM capsaicin (black) and of capsaicin after 5 min of 5 μM OA (magenta). (**c**) Remaining current fractions after OA in inside- and outside-out patches (Ctrl or controls were obtained with recording solution alone for 5 min) as normalized to the current with capsaicin (120 mV). *$P < 0.01$ between groups as compared by brackets, $n = 6$. (**d–f**) OA inhibits currents activated by different stimuli. Representative currents at ±120 mV were obtained as in **a**. (**d**) Activation by 5 μM LPA (magenta trace) was inhibited by 5 μM OA (5 min) (blue trace) in an inside-out patches. (**e**) Activation by a pH 5.5 solution in an outside-out patch (gold trace) was inhibited by 5 μM OA (blue trace). (**f**) A concentration of 5 μM cyclic phosphatic acid (cPA)-activated currents (orange trace) were inhibited by 5 μM OA (blue trace, inside-out patch). (**g**) Fraction of remaining currents activated by 5 μM LPA ($n = 10$), pH 5.5 ($n = 21$) or 5 μM cPA ($n = 5$) after exposure (5 μM OA) for 5 min as normalized to the current before OA (120 mV). (**h**) Representative traces of TRPV1 currents activated by a fast temperature ramp initially (black) and after 5 min in recording solution (magenta). *Denotes $P < 0.001$ versus the initial current. (**i**) Representative traces obtained with the same temperature ramp as in **h** initially (black) and after a 5 min incubation with 5 μM OA (magenta). (**j**) Average of experiments shown in **h** and **i**. The grey bar (control) was obtained by dividing currents after 5 min in solution recording (magenta in **h**) by the initial (black) currents and the magenta bar was obtained by dividing the remaining currents after OA (magenta trace in **i**) by the initial currents (black) $n = 7$ for both cases and *denotes $P < 0.001$ between currents inhibited by OA and the control group.

was significantly ($P < 0.0001$) reduced by the presence of capsaicin (99.6 ± 0.2% of interaction of OA with TRPV1 in the absence of capsaicin versus 42.8 ± 4.1% of interaction of OA with TRPV1 in the presence of capsaicin; Fig. 4f), which is consistent with OA competing for the same site where capsaicin binds.

To functionally evaluate whether OA binds to the VBP, we constructed a rTRPV1 channel containing mutations Y511A-S512A rTRPV1, which were previously shown to reduce capsaicin binding to rTRPV1 and abolish activation by capsaicin[21,22]. To assess whether this channel is inhibited by 5 μM OA, we used outside-out membrane patches expressing the WT rTRPV1 and the rTRPV1-Y511A-S512A mutant channels. Fig. 5a,b shows that rTRPV1 WT channels activated by pH 5.5 solutions were significantly inhibited (85 ± 2.4%) by 5 μM OA and the

inhibition of rTRPV1-Y511A-S512A channels (83.5 ± 4.8%; Fig. 5b) was not significantly different from the WT channels (85 ± 2.4%).

Another important residue affecting capsaicin binding to rTRPV1 is T550. In CkTRPV1, the equivalent of this residue is A558 (Fig. 4a). Consequently, we tested whether the mutation of this site alone or together with Y511 or with Y511-S512 could ablate OA's inhibitory effects. The T550A mutant displayed a 51 ± 2 % inhibition (Fig. 5b), whereas the Y511A-T550A mutant was inhibited 54 ± 9% (Fig. 5a, bottom left panel and Fig. 5b). The triple mutant (Y511A-S512A-T550A) yields TRPV1 channels that are only inhibited by 26 ± 10%, as compared with 85% for the WT rTRPV1 (Fig. 5a, top right panel; Fig. 5b). Accordingly, the substitution of T550 to I550 in the triple mutant also yields

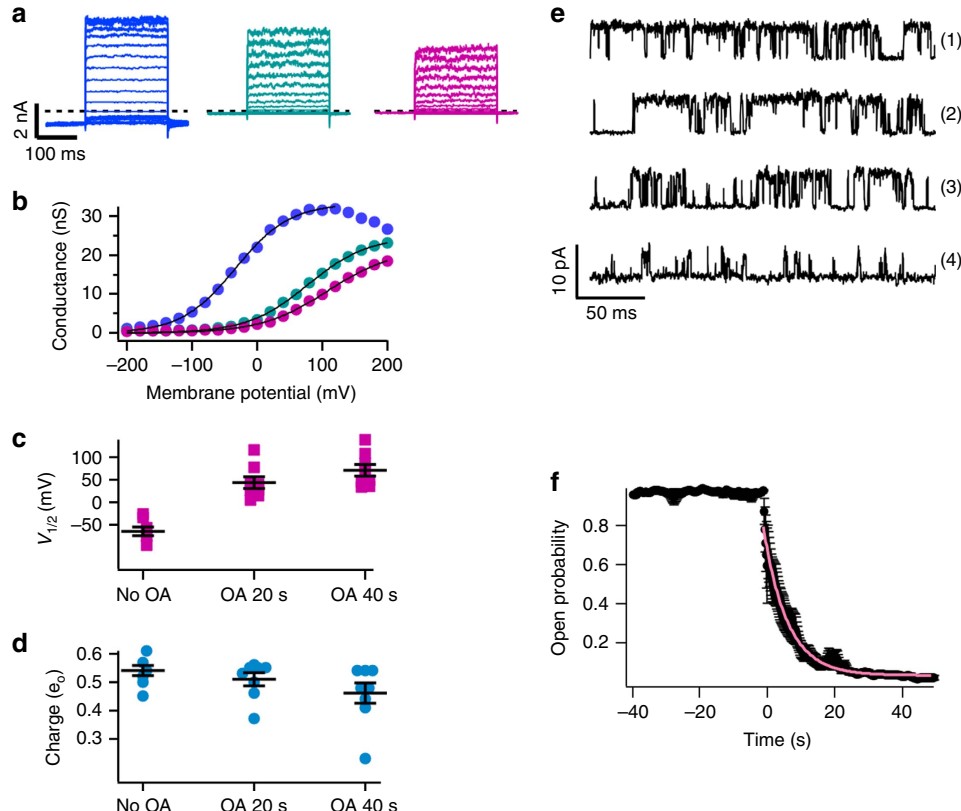

**Figure 3 | Oleic acid inhibits rTRPV1 activation by voltage and capsaicin. (a)** Currents activated by voltage (60 mV) in the presence of 4 μM capsaicin (left), and after 20 (middle) and 40 (right) s of application of 5 μM OA. **(b)** The normalized conductance obtained from currents shown in **a**. In the absence of OA (no OA), blue; 20 s after OA, turquoise; 40 s after OA, pink. A fit to equation (1) (see Methods) is shown by the black curves. **(c)** The value of the voltage of half activation, $V_{1/2}$, as a function of time after OA exposure, $n = 8$ patches. For no OA $= -63 \pm 9.2$ mV, 20 s after OA $= 44 \pm 12.9$ mV and 40 s after OA $= 73.5 \pm 13.4$ mV. **(d)** The value of the steepness of the Boltzmann fit, $q_{a}$, as a function of time after OA exposure ($n = 8$ patches). No OA $= 0.54 \pm 0.018$ $e_o$, 20 s after OA $0.51 \pm 0.23$ $e_o$ and 40 s after OA $0.46 \pm 0.37$ $e_o$. **(e)** Effect of OA on the single-channel activity of rTRPV1. Representative traces of a single-channel recording experiment at 120 mV. Traces are numbered according to: no OA, which is the activity in the presence of 4 μM capsaicin before 5 μM OA application and 6, 12 and 40 s after OA. **(f)** Average time course of the effect of OA on the single-channel open probability of TRPV1. Each point is the open probability as calculated from 1.2 s of recording ($n = 4$). Error bars are s.e.m. Time zero indicates the moment of application of OA + capsaicin. The pink curve is a fit to an exponential function with time constant 6.2 s.

TRPV1 channels that are much less sensitive to OA than the WT channels, displaying only $23 \pm 8.8\%$ of inhibition (Fig. 5b).

Finally, we tested whether introducing the A558T mutation in the CkTRPV1 ion channel (equivalent to the T550 site in the rTRPV1) could render these channels sensitive to OA. The results shown in Fig. 5a (bottom right) and Fig. 5b demonstrate that these channels become responsive to OA, since the inhibition of low pH-induced currents exhibited by this mutant is now significantly different from the WT CkTRPV1 channels, being $69 \pm 11\%$ as compared with $29 \pm 13\%$, respectively (Fig. 4b,d). Taken together, these data provide clear evidence that OA interacts within the VBP of TRPV1.

**Docking of OA in the VBP.** The experiments mentioned above are consistent with OA occupying, at least partially, the same binding site as capsaicin. To further test this hypothesis, we performed molecular docking simulations between a putative closed state of TRPV1 and OA. Using the recently published cryo-electron microscopy structure of TRPV1 (ref. 2) as a template, we found a possible basis for binding of OA to this site. In the WT channel, the VBP is accessible from the membrane surrounding the first four transmembrane segments. OA binds in the VBP in a configuration that is stabilized by a hydrogen bond

between T550 and the carbonyl group in OA. Residues Y511 and S512 seem to provide a hydrophobic environment and just small enough binding site for the long, bent hydrophobic tail of OA (Fig. 5c,d). In contrast, binding of OA to the triple mutant channel is greatly destabilized by the absence of a hydrogen bond donor at position 550. Moreover, the exchange of Y511 and S512 for the smaller alanine, allows OA to bind in configurations that lack stable interactions with the VBP (Fig. 5e).

**Physiological role of OA in inhibiting pain and itch.** To establish whether OA would also be inhibitory under more physiological conditions, we measured, in mouse dorsal root ganglia (DRG) neurons, whether OA would also suppress endogenously expressed TRPV1. Following the same protocol as described above for inside-out patches, we found that 5 μM OA inhibited the currents activated by 4 μM capsaicin by $86.6 \pm 4\%$ (Fig. 6a,b). These data suggest that OA may modulate TRPV1-evoked physiological responses.

To substantiate this hypothesis, in mice, we investigated the acute effects of OA on TRPV1-mediated pain-related behaviour produced by capsaicin and LPA by means of a standard paw-licking assay[4]. Under control conditions (injection of vehicles or 3.1 μg OA), the mice licked their paws for

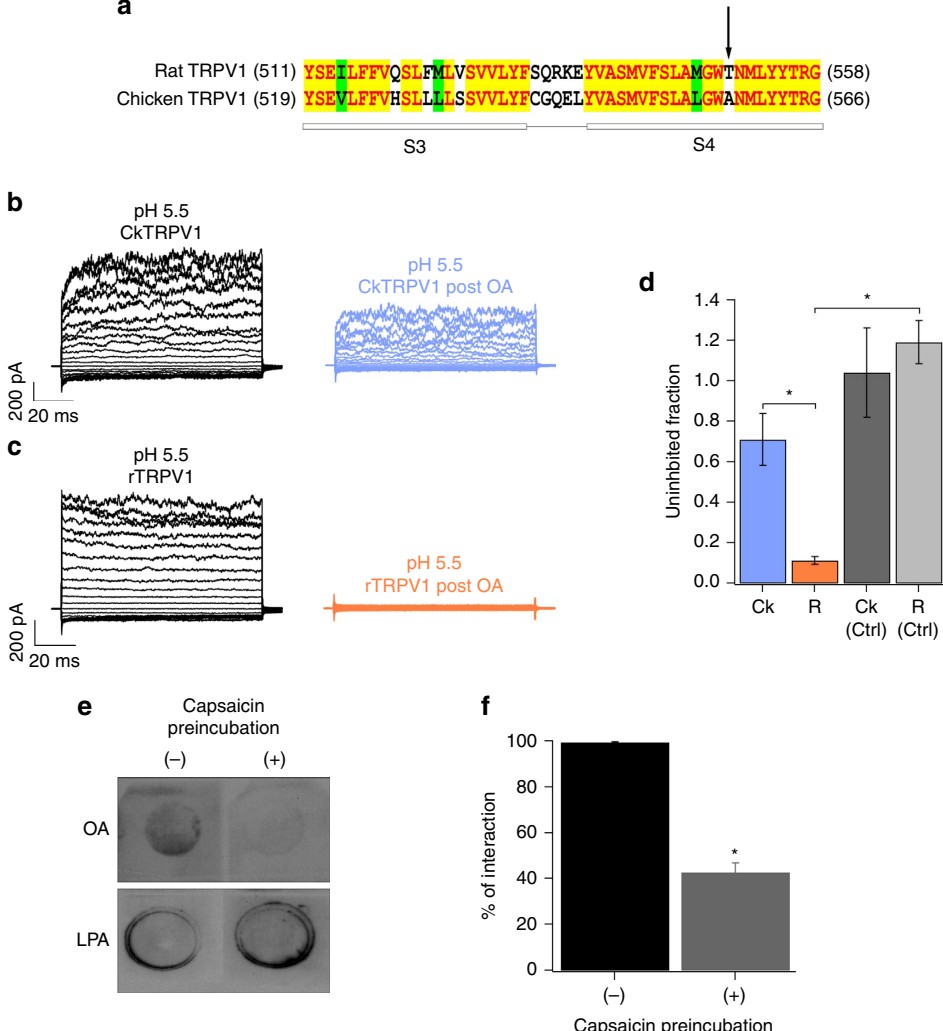

**Figure 4 | Inhibition of TRPV1 from different species by oleic acid. (a)** Sequence comparison between rat (r) and chicken (Ck) TRPV1 vanilloid-binding pockets (VBPs). Yellow denotes conserved residues. The arrow denotes residues T550 in rat and A558 in CkTRPV1. **(b,c)** Representative TRPV1 currents evoked from a pH 5.5 solution (black traces) and after 5 μM OA in an outside-out patch for CktRPV1 (blue traces) **(b)** or rTRPV1 (orange traces) **(c)** expressing HEK293 cells. **(d)** Fraction of remaining currents in CkTRPV1 (Ck) (blue bar; $n = 30$) and rTRPV1 (R) channels (orange bar; $n = 30$). Ck (Ctrl, $n = 10$) and R (Ctrl, $n = 8$) are CkTRPV1 and rTRPV1 outside-out patches exposed to vehicle (recording solution + 0.14% ethanol (EtOH) for 5 min and reactivated by pH 5.5. *$P < 0.001$ between groups as compared by brackets. Grey bars represent control experiments with vehicle solution for Ck and rTRPV1. **(e)** Overlay assay for LPA and OA with rTRPV1 protein preincubated in the absence ( − ) and presence of 0.5 μmol capsaicin ( + ). LPA (0.2 μmol) signals were not affected by capsaicin, while OA (2 μmol) signals were fainter when TRPV1 was preincubated with capsaicin (top panel). **(f)** The percentage of interaction of TRPV1 protein with OA or LPA was obtained by densitometric analysis of the spots and normalized to the positive control, which was LPA itself. For OA in the absence of capsaicin ( − ) per cent of interaction was 99.6 ± 0.2 and in the presence of capsaicin ( + ) per cent of interaction was 42.8 ± 4.1, $n = 6$ for all groups. *$P < 0.0001$, $t$-test.

~15 s (Fig. 6c; Supplementary Table 2). Paw-licking times when the different vehicles and OA were injected, displayed no significant differences among them (Fig. 6c; Supplementary Table 2), showing that OA, by itself, does not affect other pain-producing pathways, at least those associated with paw-licking. However, as compared with controls, when mice were injected with 0.75 μg of capsaicin or 4.1 μg of LPA, they displayed statistically significant increases in paw-licking responses (39.4 ± 2.1 s for capsaicin and 43 ± 1.6 s for LPA). Finally, when 3.1 μg of OA was injected together with capsaicin or LPA, these licking responses were significantly reduced to 24.4 ± 2 s and 27.2 ± 2.9 s, respectively (Fig. 6c; Supplementary Table 2).

We then tested weather OA could also modulate another physiological process, namely, itch, which partially depends upon

the activation of TRPV1 (refs 9,23–25). During experiments to determine the effects of cPA in pain assays, we noticed that the mice began to scratch. To determine whether cPA was responsible for this behaviour, and whether this behaviour could be modulated by OA, we injected control solutions (vehicle or 2 μg g$^{-1}$ of weight of OA) and 1.7 μg g$^{-1}$ of weight of cPA with and without OA into to the necks of C57BL/6J WT mice and quantified the scratching bouts (Fig. 6d; Supplementary Table 3). These data show that injections of the vehicle or OA produced small or negligible scratching responses (11.6 ± 1 and 7.3 ± 1.6 bouts of scratching for vehicle and OA, respectively; Fig. 6d; Supplementary Table 3). In contrast, after the injection of cPA, mice exhibited a large increase in scratching bouts (31.6 ± 2.7) that was significantly inhibited (17.3 ± 3.2) by OA (Fig. 6d; Supplementary Table 3). To verify that this itch response

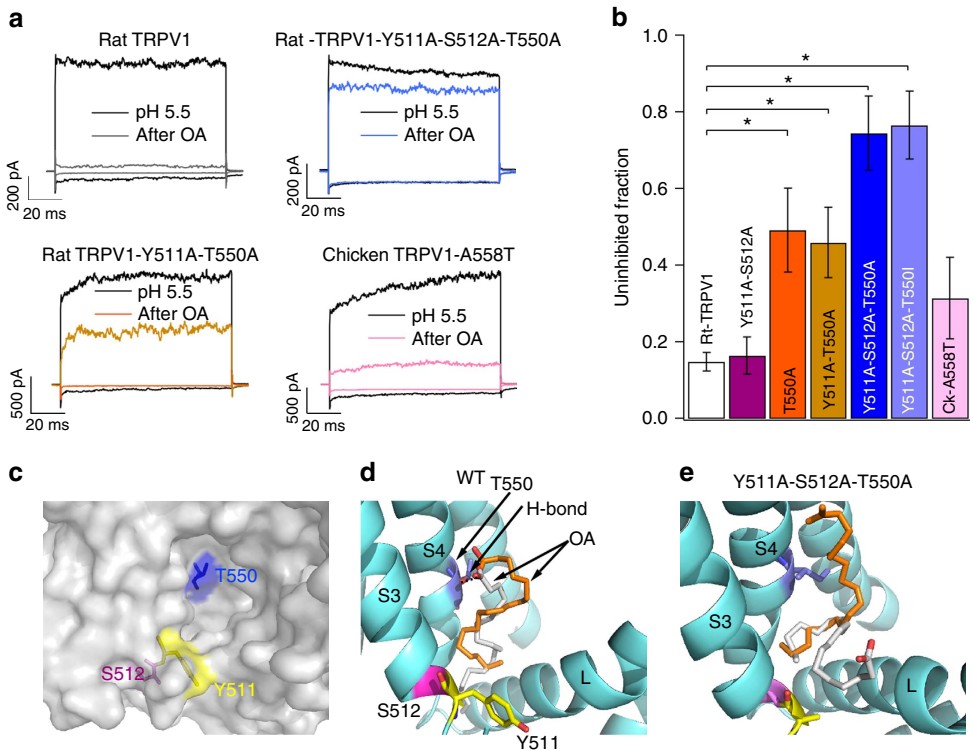

**Figure 5 | Oleic acid binds to the VBP of TRPV1. (a)** Representative traces obtained from outside-out membrane patches as in Fig. 2. Currents are from WT rTRPV1, rTRPV1-Y511A-T550A, rTRPV1-Y511A-S512A-T550A and CkTRPV1-A558T channels. Black traces were obtained in response to pH 5.5, and turquoise, gold, blue and pink traces were obtained in the presence of pH 5.5 after 3 min of 5 μM OA. **(b)** Fraction of remaining currents after OA as normalized to the initial current in the presence of pH 5.5; rTRPV1 ($n = 22$); rTRPV1-Y511A-S512A ($n = 10$); rTRPV1-T550A ($n = 14$); rTRPV1-Y511A-T550A ($n = 8$); rTRPV1-Y511A-S512A-T550A ($n = 9$), rTRPV1-Y511A-S512A-T550I ($n = 11$) and CkTRPV1-A558T ($n = 15$) *$P < 0.001$ between groups as compared by brackets. **(c)** Surface representation of vanilloid-binding site (VBP) in the closed state structure (PDB 3J5R). A cavity is visible and the location of mutated residues is indicated: T550 (blue), Y511 (yellow) and S512 (magenta). **(d)** Docking calculation showing two representative protonated OA configurations with the lowest energy and highest occupancy binding to the VBP. Both conformations have $-5.2$ kcal mol$^{-1}$ of binding energy. The average distance from the carbonyl oxygen in OA to the hydroxyl group in T550 is 2.5 Å, allowing for good hydrogen bonding (black dotted line, see arrow) between T550 and the carbonyl in OA. **(e)** Docking of OA to the triple mutant rTRPV1-Y511A-S512A-T550A. Mutations Y511A-S512A increase the size of the VBP and in combination with the absence of the hydrogen bond between OA and T550, reduce the binding energy to $-4$ kcal mol$^{-1}$, allowing OA to bind in a more disordered conformation. Labels correspond to the S3, S4 and S4–S5 linker helices.

was due, at least in part, to TRPV1, the experiment was repeated using $Trpv1^{-/-}$ mice. As shown in Fig. 6d, the injection of cPA elicited a scratch response that was less prominent in $Trpv1^{-/-}$ mice ($11.1 \pm 2.0$ bouts of scratching) than the one observed in their WT littermates and similar to the response elicited by the injection of the vehicle in the $Trpv1^{-/-}$ mice ($11.6 \pm 2$ scratching bouts). Finally, we tested whether OA would inhibit the itching effects of histamine, whose actions partially depend on the activation of TRPV1 (ref. 9). This was accomplished by injecting 50 μg histamine, which induced an increase in the number of scratching bouts as compared with mice injected only with vehicle ($87.7 \pm 5.5$ versus $11.6 \pm 1$, respectively; Supplementary Fig. 6). When PA ($1.25 \mu g g^{-1}$ of weight), a fatty acid that did not inhibit activation of TRPV1 by capsaicin (Supplementary Fig. 3) was co-injected with histamine, there were no significant differences with respect to histamine alone ($80.6 \pm 15$). However, when histamine was co-injected with OA ($1.25 \mu g g^{-1}$ of weight), a reduction of 58% in the response to histamine was observed (Supplementary Fig. 6), indicating that OA specifically decreases histamine-dependent itch.

To definitely distinguish between itch- and pain-related behaviours, we performed experiments in mice using the 'cheek model'[26]. The pain-related component was elucidated by injecting 0.1 μg capsaicin that produced an increase in the

amount of wiping bouts that the mice displayed using their forelimbs, as compared with animals injected only with vehicle ($92.7 \pm 10.3$ versus $25.2 \pm 7.6$, respectively; Supplementary Fig. 7a). As a control, we evaluated the effects of the injection of 5 μg of OA on pain-related behaviour, which resulted in negligible wiping responses ($7.6 \pm 3.7$; Supplementary Fig. 7a). However, when OA (5 μg) was co-injected with capsaicin (0.1 μg), the pain-related responses were significantly reduced as compared with animals injected only with capsaicin ($43.9 \pm 4.8$ versus $92.7 \pm 10.3$, respectively; Supplementary Fig. 7a). When we evaluated the itch-related response using this same cheek model, as evidenced by scratching with the hind limb, we found that injection of capsaicin does not produce a significant increase in scratching bouts as compared with injection with vehicle alone ($5.3 \pm 1.1$ versus $8 \pm 5.7$, respectively; Supplementary Fig. 7b). Moreover, injection of OA and OA + capsaicin did not produce an increase in itch-related behaviour ($4.2 \pm 2$ versus $6 \pm 2.9$, respectively; Supplementary Fig. 7b). We also performed experiments to determine the effects of OA on histamine-induced pain or itch in the cheek model. The results show that, while there are no significant differences in pain-related behaviour between animals injected with histamine (30 μg) and the vehicle-injected group ($15 \pm 6.3$ versus $12 \pm 6.4$, respectively; Supplementary Fig. 7c), injection of histamine produced a

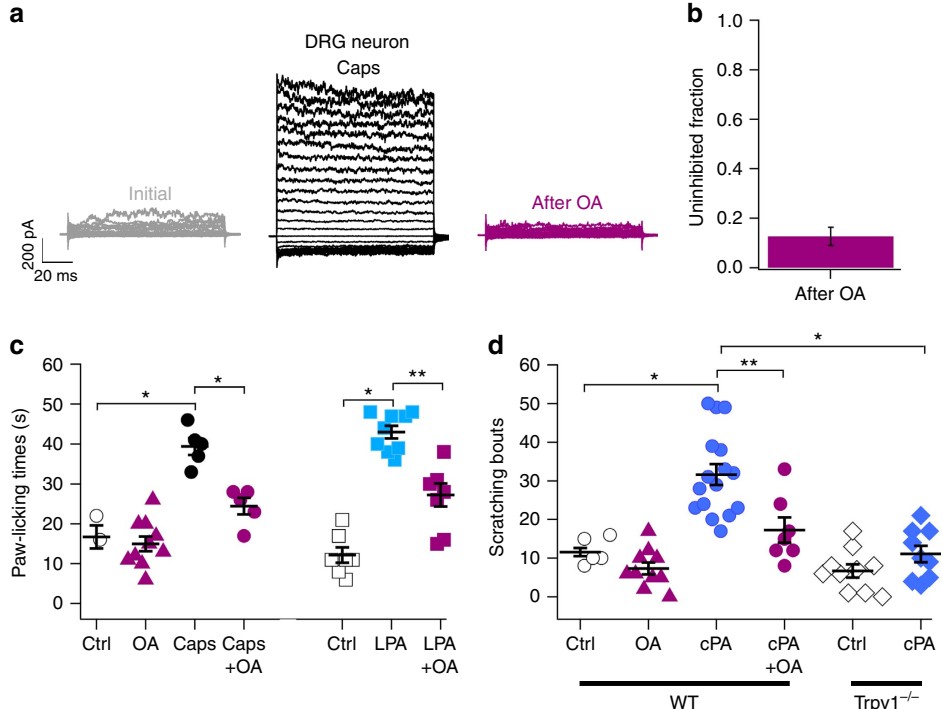

**Figure 6 | Oleic acid attenuates painful and pruritic behaviours. (a)** Representative TRPV1 currents (as obtained as in Fig. 2) from an inside-out patch from DRG neuron before (grey traces) and after exposure to 4 μM capsaicin (black traces). After wash, 5 μM OA was added for 5 min and the patch was re-exposed to capsaicin (magenta traces). **(b)** The fraction of uninhibited currents was 13 ± 3.6%, as obtained by normalizing the data obtained in the presence of capsaicin after OA to the data obtained with capsaicin before OA ($n = 9$). **(c)** Average paw-licking times were: 16.7 ± 2.9 s for capsaicin and 12.1 ± 1.9 s for LPA vehicle injections (Ctrl; $n = 3$–7); 15.0 ± 1.8 s for OA ($n = 10$); 39.4 ± 2.1 for capsaicin ($n = 5$) and 43 ± 1.6 s for LPA ($n = 9$); 24.4 ± 2 s for Caps + OA ($n = 5$) and 27.2 ± 2.9 s for LPA + OA ($n = 8$). *$P < 0.001$ and **$P < 0.0001$ between groups as compared by the brackets. Also see Supplementary Table 2. **(d)** The average number of scratching bouts were: 11.6 ± 1.1 for vehicle-injected (Ctrl; $n = 10$); 7.3 ± 1.6 for OA ($n = 10$); 31.6 ± 2.7 for cPA ($n = 16$); 17.3 ± 3.2 for cPA + OA ($n = 7$) for WT animals and 6.7 ± 1.7 for control ($n = 10$) and 11.1 ± 2.1 for cPA-injected ($n = 9$) $Trpv1^{-/-}$ mice. For WT mice: *$P < 0.0001$ and **$P < 0.01$ between groups as compared by the brackets. Also see Supplementary Table 3.

significant increase in scratching bouts (itch) when compared with injection of vehicle (70.8 ± 10.2 versus 5.2 ± 3, respectively, Supplementary Fig. 7d). Finally, when histamine was co-injected with OA (30 and 10 μg, respectively), the number of scratching bouts was significantly decreased in comparison with injection of histamine alone (34 ± 6.4 versus 70.8 ± 10.2, respectively; Supplementary Fig. 7d).

These results are in agreement with what we obtained using the paw-licking or the neck-scratching assays.

## Discussion

TRPV1 channels are versatile molecules whose activation has been linked to several important physiological and pathophysiological processes[27]. Here we have shown that OA is a naturally occurring antagonist of this important channel and is physiologically relevant in that, by inhibiting TRPV1 in peripheral neurons, it partially blocks pain evoked by its agonists and itch evoked by histamine and a novel pruritic agent, cPA.

Although OA has been shown to modulate other types of ion channels[28–34], its molecular mechanisms have not been elucidated. We found that OA acts as an allosteric inhibitor of TRPV1 that stabilizes a closed state (Fig. 3e,f), increases the voltage necessary to activate the channel (Fig. 3a–d) and inhibits responses to acid, capsaicin, LPA and elevated temperature (Fig. 2), all of which promote channel activation by interacting at different sites[3,4]. The structural requirements for fatty acids to inhibit TRPV1 are primarily dependent on the presence of an

unsaturation and on the length of the acyl chain (C18 *cis* 9; Supplementary Fig. 3). We also found that LA (C18 *cis* 9,12) could inhibit TRPV1-mediated currents, albeit more moderately than an equal OA concentration. The polar group is also important since molecules with the same acyl chain as OA (C18:1), such as oleoylethanolamide[35], 1-monoglyceride (18:1–3)[36], olvanil (which binds to the VBP)[37] and LPA[4] all activate TRPV1. Thus, small differences can determine whether a molecule with a similar structure to OA is a TRPVI activator, an inhibitor or has no effect.

Two recent studies described capsaicin binding to the VBP[19–21]. In both, capsaicin is oriented with the acyl chain pointing out of the VBP, almost parallel to the transmembrane domains and with the polar head occupying a cavity between the S3 and S4 transmembrane segments and the S4–S5 linker. Residue T551 of the hTRPV1 (T550 in rTRPV1) is proposed to interact with the amide oxygen in capsaicin[21]. These studies also found residues Y511 and S512 to be important for capsaicin binding in the VBP. Given that the OA head group is much simpler than the aromatic head group, amide and ester regions of capsaicin, it is perhaps not unexpected that the predicted orientation of OA may differ from that of capsaicin, in that the acyl chain occupies the cavity, while the acidic region forms hydrogen bonds with residues outside the cavity. In our study, amino-acid residue T550 is clearly important for the binding of OA to TRPV1, presumably by forming hydrogen bonds with the carbonyl group of OA, as suggested by our mutagenesis experiments and docking simulations (Fig. 5). Nevertheless, the

presence of residues Y511 and S512 is also important for OA binding, as they may contribute to the conformational constraints of the acyl chains. For example, fatty acids with shorter acyl chains might occupy the cavity, but the acid group would be unable to reach residue T550 to form a hydrogen bond, thus weakening the interaction. We note that OA is protonated in the VBP, since in a low-dielectric media such as the VBP the $pK_a$ of OA will increase[38].

Most recently, Gao et al.[39] obtained a TRPV1 structure by incorporating this protein to lipid nanodiscs. In the apo structure of TRPV1, the authors report the presence of a lipid in the VBP, which they interpret as a phosphatidylinositol that is hypothesized to bind to a closed state and would be displaced by diverse activators that bind to the VBP[39].

The fact that OA can inhibit activation by acidic solutions, capsaicin, LPA and temperature indicates that once bound, OA promotes a conformational change that stabilizes the channel in a long-lived closed state. The observed preference of OA for the closed state may provide functional selectivity[40] and, along with its effects on activation by voltage, an allosteric mechanism of inhibition that shifts the gating equilibrium to a closed conformation, is also consistent with our results. Gavva et al.[41] had previously proposed a classification of TRPV1 antagonists based on their ability to inhibit the activation of these channels by capsaicin and protons (group A) or on their ability to inhibit activation by capsaicin but not protons (group B). These authors stated that group A antagonists lock the TRPV1 channels in the closed state, which is consistent with what we find with OA and TRPV1.

It is important to mention that Matta et al.[42] have previously tested OA on TRPV1 activity. Using Xenopus oocytes and performing voltage-clamp experiments, they found that 50 μM OA exhibited small antagonistic effects on capsaicin activation. Differences between our study in which the response to capsaicin was inhibited 85% (Fig. 2) and this other study by Matta et al.[42], could arise from several factors including: OA preparation and concentration, the application time and a different expression system. We note that in this same study, the authors showed that docosahexaenoic, eicosapentaenoic and linolenic (LNA) at a concentration of 10 μM evoked TRPV1-mediated currents[42], rather than inhibiting the activity of this channel.

A question arises as to what role OA may have under physiological/pathological conditions? In mammalian astrocytes, albumin stimulates OA synthesis that promotes neuronal differentiation and serves as a neurotrophic factor[43]. Both albumin and OA can be incorporated into neurons and lead to dendritic growth through the activation of the peroxisome proliferator-activated receptor α (PPARα) receptor[44]. Moreover, the albumin–OA complex has been actually proposed for the treatment of paralysis, spasticity and pain[45]. With regard to OA's concentrations, under physiological conditions it was found that, in human blood, these range between 10 and 100 μM (ref. 46), and these concentrations increase under ischaemic conditions[47].

There are only few reports that describe the effects of OA on the biophysical properties of ion channels. While $K_{ACh}$ channels are inhibited by 5 μM OA and $BK_{Ca}$ channels are activated by 10 μM (ref. 28), Kv7.2/3 channels remain unaffected by 70 μM OA[29]. For sodium and calcium channels in CA1 neurons[30] and for potassium channels in taste cells[31], 16 μM and 10 OA μM, respectively, have little or no effect. For human cardiac Nav1 channels and gap junction channels in smooth muscle cells, 5 μM and 25 μM OA, respectively, display inhibitory effects[32,33]. In human skeletal muscle, sodium channel currents (hSkM1) are increased by 5 μM OA[34]. Finally, in trigeminal neurons 30 μM OA increases $Ca_i^{2+}$ influx[48].

Our physiological studies with mice showed that OA partially inhibited pain responses produced by both capsaicin and LPA (Fig. 6; Supplementary Fig. 7). Importantly, we identified cPA, a novel, TRPV1-dependent itch-causing agent, whose actions were abolished in the $Trpv1^{-/-}$ animals and by OA in WT animals. One of the pathways associated with TRPV1 and itch generation involves by-products of the lipooxygenase pathway activated by histamine receptors[10]. Our experiments show that OA also partially inhibits histamine-induced itch (Fig. 6; Supplementary Fig. 7), showing that it is an effective negative regulator of pruritus. In summary, in this study, we identify OA as a new and natural inhibitor of TRPV1 and provide a basis for understanding its effects on reducing both pain and itch.

## Methods

**DRG and HEK293 cell culture and recording.** In brief, DRG neurons were isolated by manual dissection. C57BL/6J mice were anaesthetized with halothane, decapitated and their spinal columns were removed. The spinal column was bisected and whole ganglia were excised from the surrounding tissue into DMEM (Invitrogen). Ganglia were digested with collagenase 4 mg ml$^{-1}$ and 1 mg ml$^{-1}$ of trypsin in DMEM. Neurons were dissociated from digested ganglia by manual trituration with a fire-polished, serum-coated glass pipette. Finally, neurons were resuspended in DMEM medium + 10% fetal bovine serum (HyClone) and plated in a small volume onto glass coverslips coated with 1 mg ml$^{-1}$ poly-D-lysine (Sigma-Aldrich). After 5 h, the neurons were immersed in fresh medium with 100 ng of 2.5S nerve growth factor (NGF) (Merck-Millipore). Neurons were used for electrophysiology between 12 h after dissection[4]. HEK293 cells (American Type Culture Collection) were transfected either with WT or mutant rat TRPV1 (rTRPV1), chicken TRPV1 (CkTRPV1), human TRPV1 (hTRPV1) cloned in the pcDNA3 plasmid and in conjunction with pIRES-GFP (BD Biosciences) using JetPEI transfection reagent (Polyplus Transfection). Currents were recorded using the inside-out and outside-out configurations of the patch-clamp technique[49]. A high-resistance seal was obtained by pressing a borosilicate glass (Sutter Instruments Company) micropipette against the cell membrane and suction was applied. Inside-out membrane patches were obtained by pulling the micropipette was withdrawn from the cell, while for outside-out patches, more suction was applied to rupture the membrane patch first and then micropipette was withdrawn from the cell.

**Solution preparations.** Inside-out and outside-out recordings were performed under isotonic conditions with the following solutions (mM, Sigma-Aldrich): 130 NaCl, 3 HEPES (pH 7.2) and 1 EDTA. For experiments where low pH (5.5) solutions were applied to the extracellular surface of the TRPV1 channel in outside-out membrane patches, HEPES was substituted by 3 mM MES (Sigma-Aldrich). Stocks for capsaicin (Sigma-Aldrich) were prepared in ethanol; LPA (Avanti Polar Lipids) was prepared as previously described[4] and cPA (Avanti Polar Lipids) stocks were prepared in the same way as LPA. OA, stearic acid, ricinoleic acid, PA, LA and LNA were all purchased from Sigma-Aldrich, aliquoted and then prepared as a 3.5 mM stock in ethanol (J.T. Baker) by vortexing for 2 min followed by a 5 min bath sonication.

OA (C18:1 cis 9), hexadecylglycerol (1-O-hexadecyl-sn-glycerol), nervonic (C24:1 cis 9), petroselinic (C18:1 cis 6), elaidic (C18:1 trans 9), erucic (C22:1 cis 13, omega-9), ricinoleic (12-hydroxy-9-cis-octadecenoic), stearic (C18:0), PA (C16:1 cis 9 omega 7), LA (C18:2 cis, cis 9,12) and LNA (C18:3 cis, cis, cis 9,12,15) acids, all from Sigma-Aldrich, were directly prepared as stock solutions of 3.5 mM in ethanol. All stocks except for OA were frozen to −70 °C. Before use, they were bath-sonicated for 5 min using a Branson 1510 ultrasonicator (40 kHz). With the exception of OA, for which a dose–response curve was generated, all these others were tested at 5 μM.

All stocks were diluted to the desired concentration in the recording solution. Solutions were changed with an RSC-200 rapid solution changer (Bio-Logic Science Instruments). Perfusion lines were replaced every 2 days to avoid lipid binding to Teflon tubing and undesired fluctuations in compound concentration. Experiments were performed at 24 °C. Currents were recorded using voltage protocols with a holding potential of 0 mV and steps from −120 to 120 mV applied for 100 ms and then the voltage was returned to 0 mV. Capsaicin and pH 5.5 solutions were applied for 10 s, and LPA and cPA for 5 min at the end of which currents were measured with the voltage protocol described above. The currents of channels exposed to OA were normalized to the maximal current activated by a given agonist. The time course for recovery from inhibition was obtained by holding the patches at +40 mV for 350 ms and by applying capsaicin, washing it off with recording solution, applying capsaicin + OA and finally assessing the time course with which the current recovered in the presence of capsaicin alone. Time courses of inhibition by 5 μM OA were obtained using continuous pulses at −60 and 60 mV. For the open state experiment, capsaicin and OA were co-applied and fits from independent experiments were averaged. For the closed state, each time point was obtained by averaging several patches, exposed to OA in the absence of

capsaicin. The decay in the current was fitted to a single exponential to obtain the time course of inhibition ($\tau$).

Experiments to show that OA does not activate TRPV1 were obtained by applying continuous pulses to $+40$ and $-40$ mV for 600 ms and activating TRPV1 with capsaicin, removing it with recording solution and then applying 5 µM OA alone for 3 min to show that no currents were activated in the presence of the fatty acid (Supplementary Fig. 1). After 3 min with OA, the inhibition was assessed by exposing the patches to capsaicin. Moreover, in patches held at $+40$ mV, we washed the capsaicin-activated remaining currents obtained after the first incubation with OA and reapplied OA for another 2 min to determine that OA did not in fact activate TRPV1 currents by itself and then reactivated them with capsaicin alone.

**Voltage- and temperature-dependence experiments.** Changes in the voltage-dependent activation of TRPV1 in the presence of capsaicin were assessed by applying a voltage protocol to inside-out patches, which were held at $-60$ mV for 50 ms and then the voltage was stepped from $-200$ to 200 mV for 200 ms and back to $-60$ mV for 10 ms. This protocol was applied before and after every 20 s of exposure to 5 µM OA. Conductance was calculated according to Ohm's law and the normalized conductance fitted to a Boltzmann equation.

$$G/G_{max} = \frac{1}{1 + \exp^{-q_a(V - V_{1/2})/kT}} \quad (1)$$

Here $G_{max}$ is the maximal conductance, $q_a$ is the charge associated with voltage activation, $V_{1/2}$ is the voltage, where $G/G_{max}$ is 0.5 and $kT$ has its usual meaning.

Currents were low-pass filtered at 2 kHz and sampled at 10 kHz with an EPC 10 amplifier (HEKA Elektronik) and were plotted and analysed with Igor Pro (Wavemetrics Inc.).

To study the effects of OA of activation of TRPV1 by temperature, we used a previously described method[11]. In brief, temperature ramps from 22.3 to 44.8 °C were generated with a resistive microheater, and applied before and after the application of 5 µM OA to TRPV1-expressing HEK293 membrane patches. We first determined that TRPV1 currents activated in response to temperature ramps were not substantially diminished by a subsequent ramp after 5 min in recording solution and then we performed experiments where the initial activation by temperature was compared with that obtained after a 5 min incubation with 5 µM OA.

**Dose–response and inhibition by OA in the closed and open states.** Dose–response curves for the inhibition of capsaicin-induced TRPV1 currents were performed by applying a given concentration of OA in the absence of capsaicin to inside-out excised membrane patches of HEK293 cells expressing rTRPV1 channels. A single concentration of OA was tested in each membrane patch and the data of several patches at a given concentration were pooled. All currents were measured at a voltage of $+120$ mV. Data were normalized to the currents initially obtained in the presence of only 4 µM capsaicin. The Hill equation was fitted to the data as previously described[4].

$$\frac{I}{I_{max}} = \left(\frac{1}{1 + \frac{[OA]}{[K_D]}}\right)^n \quad (2)$$

Experiments to determine state dependence of inhibition of TRPV1 by OA were performed by first measuring the currents in the presence of a given agonist and then applying OA in the presence (open state) or absence (closed state) of the agonist and then re-measuring the currents in the presence of only the agonist. Dose–responses in the presence of different capsaicin concentrations were obtained at $+120$ mV. A half-saturating concentration of OA was applied either in the presence of a given capsaicin concentration or capsaicin was washed and OA was applied in the absence of the agonist and then currents were re-measured at a given capsaicin concentration. The data were normalized to the currents initially obtained in the presence of 4 µM capsaicin and also fitted to the Hill equation.

**Single-channel recording and analysis.** The effect of OA on single TRPV1 channel kinetics was studied in inside-out patches[15,16]. Recording pipettes had a resistance $>10$ MΩ. Current was filtered at 2 kHz and sampled at 5 kHz. Patches containing only one channel activated by 4 µM capsaicin were identified as those that did not contain overlapping opening events. Current was continuously sampled for periods lasting 1.2 s at 60 mV. Single-channel openings and closures were identified with the half-threshold crossing technique[50]. The channel open probability was calculated for each sweep as the sum of the total open time divided by the sweep duration. Dwell times in the closed or open states were collected in logarithmic time histograms according to the Sine–Sigworth transformation[51]. Sums of three or two exponential components were fitted to histograms using a least-squares algorithm.

**Mutagenesis.** Mutations in various regions of the rTRPV1 channel were constructed using a method involving oligonucleotides synthesized to contain a mutation in combination with WT oligonucleotides in PCR amplifications of fragments of the complementary DNA. The product of the PCR reaction was then cut with two different restriction enzymes to generate a cassette containing the mutation. The cassette was ligated into the channel complementary DNA cut with the same two restriction enzymes. After transformation of bacteria with the ligation product, single isolates were chosen, and the entire region of the amplified cassette was sequenced to confirm for the mutation and ensure against second-site mutations[52].

**In vitro interaction assays.** Surface proteins were obtained from HEK293 cells transiently expressing TRPV1 channels using the Pierce Cell Surface Isolation kit (Pierce Biotechnology, Rockford, IL) following the manufacturer's instructions. Overlay assays were performed as previously described[5]. In brief, OA was spotted (2 µmol per spot) onto a nitrocellulose membrane (GE Healthcare, Pittsburgh, PA) previously blocked with 1% fatty acid-free bovine serum albumin (BSA; Calbiochem) in PBS and membranes were incubated with the surface protein solutions and exposed to anti-TRPV1 antibody (P-19, Santa Cruz Biotechnology Inc.) diluted 1:1,000 in 3% fat-free dried milk in PBST (with 0.1% of Tween). Finally, the membranes were incubated with horseradish peroxidase-conjugated secondary anti-goat antibody (sc-2033, Santa Cruz Biotechnology Inc.) diluted 1:5,000 in 3% fat-free dried milk in PBST. The binding TRPV1 to the lipid-containing spots was visualized by chemiluminescence by exposing the blot for 15 min (Amersham Bioscience, Piscataway, NJ).

For competition assays, the TRPV1 proteins were initially incubated with 0.5 µmol of capsaicin or vehicle (ethanol) and then applied to the nitrocellulose membranes that contained the OA spots. Semi-quantitative densitometric analysis was done using Image J software (NIH) and expressed as relative protein levels versus the amount of TRPV1 bound to each spot with respect to the positive control (LPA; 0.2 µmol).

**Molecular modelling and docking.** The structure of TRPV in the putative closed state (PDB accession number 3J5R) was minimized by a 200 ns cycle of molecular dynamics (CHARM force field) in the Miztli Supercomputer at UNAM. The triple mutant structure was constructed on the frame of the minimized WT model using UCSD-Chimera.

Docking of OA was carried out with Autodock 4 (ref. 53). The docking grid was centred on the VBP[21]. The bound OA structures reported in Fig. 5 are members of the smallest and most populated binding energy clusters of binding conformations calculated by Autodock. These energies are reported in the figure legend. Figures were prepared with PyMol software.

**Mice strains.** C57BL/6J WT mice were obtained from the local animal facility at the Instituto de Fisiología Celular of the National Autonomous University of Mexico (UNAM) and C57BL/6J $Trpv1^{-/-}$ mice were obtained from The Jackson Laboratory (Bar Harbor, ME). Animals were handled according to institutional standards from the US National Institutes of Health and from the local Institutional Committee (Comité Institucional para el Cuidado y Uso de Animales de Laboratorio) at the Instituto de Fisiología Celular (CICUAL TRE17–14).

**Behavioural assays.** For paw-licking assays, the injected vehicle or control solution for mice, consisted of 10 µl of either saline solution (0.9% NaCl) with DMEM + fatty acid-free BSA (1%; Calbiochem) and 0.3% of ethanol for capsaicin controls or saline solution with 1% DMEM for LPA controls. The test solutions contained OA (3.1 µg), capsaicin (0.75 µg), capsaicin + OA (3.1 µg), LPA (4.1 µg) or LPA + OA. Paw-licking behaviour was quantified for 10 min immediately after injection with a 30 G needle of all of the above combinations. For itch-related behaviour induced by cPA, vehicle 50 µl of DMEM and fatty acid-free BSA (1%), 1.7 µg of cPA for gram of weight of the animal or cPA + OA ($2 \mu g\,g^{-1}$ of weight) in 50 µl of vehicle (injected into the neck intradermally) and bouts of scratching were counted for 20 min. For cPA, experiments were performed with C57BL/6J WT and C57BL/6J $Trpv1^{-/-}$ mice.

For histamine-induced itch behaviour, C57BL/6J mice (8 weeks of age) were used. An amount of 50 µg of histamine, in a final volume of 50 µl vehicle (DMEM with 1% of BSA), was intradermally injected into the neck of the mice. After histamine injection, bouts of scratching were counted for 20 min. To evaluate the inhibitory effects of OA on itch behaviour, mice were co-injected with a total of 50 µl of histamine together with OA or phosphatidic acid (PA; $1.25 \mu g\,g^{-1}$ of weight).

We also used a method previously described to differentiate pain and itch in rodents[26]. In brief, to perform the cheek model assays, a volume of 10 µl of each chemical or their vehicles were injected intradermally into the cheeks of C57BL/6J mice (8 weeks of age) using a 0.5 ml insulin syringe with a 30 G needle. Experimental groups received either 0.1 µg of capsaicin or a dispersion of capsaicin and OA (0.1 and 5 µg, respectively) or the vehicle (DMEM + 1% of BSA) and 30 µg of histamine or histamine plus OA (30 and 10 µg, respectively) and the vehicle. All working solutions were prepared in DMEM with 1% of BSA. The number of times that mice wiped their cheeks (pain behaviour) and the number of scratching bouts (itch behaviour) were scored during a period of 20 min.

**Statistical analysis.** Statistical comparisons were made with the Student's $T$-test. A value of $P < 0.01$ was considered statistically significant. Group data are reported as the mean ± s.e.m.

**Data availability.** The data that support the findings in this study are available from the corresponding author on request.

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

## Acknowledgements

We thank Sharona Gordon for the kind gift of mutant rTRPV1-K710A, K714A, R717A and K718A, and Sebastián Brauchi for providing us with chicken TRPV1. We also thank Enrique Hernández-García for support with some single-channel experiments and Laura Ongay, Ana Escalante, Francisco Pérez, Claudia Verónica Rivera Cerecedo and Héctor Alfonso Malagón Rivero at Instituto de Fisiología Celular, UNAM for expert technical support. This work was supported by grants from Dirección General de Asuntos del Personal Académico (DGAPA)-Programa de Apoyo a Proyectos de Investigación e Inovación Tecnológica (PAPIIT) IN200314, Consejo Nacional de Ciencia y Tecnología (CONACyT) CB-2014-01-238399 and a grant from the Marcos Moshinsky foundation to

T.R.; DGAPA-PAPIIT IN209515 and CONACyT 248499 to L.D.I. and a grant from Fronteras en la Ciencia no. 77 from CONACyT to T.R. and L.D.I.; DGAPA-PAPIIT IA202815 and Beca L'Oréal-UNESCO-CONACyT-AMC to S.L.M.-L. Molecular dynamics simulations were carried out in the Miztli Supercomputer cluster at DGTIC-UNAM (user project no. SC15-1-IR-96).

## Author contributions

T.R., L.D.I., F.S.-R., A.E.L.-R. and B.S.-F. performed the electrophysiological experiments. I.L. carried out all site-directed mutagenesis and some electrophysiological experiments. S.L.M.-L. and M.O.-R. performed all animal mattings, genotyping, behaviour, biochemical assays and data analysis. T.R. performed single-channel experiments and L.D.I. performed single-channel analysis. L.D.I. performed docking and molecular dynamics simulations. L.D.I., S.A.S. and T.R. jointly conceived the study, performed the analysis and wrote the paper.

## Additional information

**Competing financial interests:** The authors declare no competing financial interests.

