## [Peer review file · Nature Communications]

Reviewers' comments:

Reviewer #1 (Remarks to the Author):

A. Summary of the key results

This exhaustive study by Morales-Lázaro and colleagues addresses inhibition of the TRPV1 ion channel, a key player in the sensation of heat, pain, and itch, by a naturally occurring lipid compound. In particular, the authors report

- (i) discovery of oleic acid (OA) as a highly specific and effective inhibitor of the TRPV1 ion channel
- (ii) identification of the mechanism of OA-inhibition as an allosteric stabilization of the closed state, sufficient to counteract activation by various stimulatory signals including capsaicin, acidic pH, lysophosphatidic acid, and membrane depolarization
- (iii) characterization of structural determinants required for the inhibitory effect of OA
- (iv) functional overlap between the binding sites of OA and capsaicin revealed by a competitive binding assay
- (v) identification of residues important for OA binding to the channel protein,
- (vi) efficacy of OA to suppress TRPV1 currents in DRG neurons under physiological conditions
- (vii) efficacy of OA to suppress in vivo pain and itch responses evoked by TRPV1 agonists.

B. Originality and interest: if not novel, please give references

This group has largely contributed to the identification of natural TRPV1 activators in the past. Thus, in addition to vanilloids, acidic pH, heat, and voltage, they have identified several lipid activators. However, in contrast to the large repertoire of known natural activators, only few natural TRPV1 inhibitors are known. In addition to a better understanding of physiological processes, discovering TRPV1 inhibitors might have clinical relevance for the treatment of pain and itch. The discovery of OA as a TRPV1 inhibitor is therefore of considerable interest, and is likely to appeal to a broad readership. I recommend the paper for publication following minor revision.

C. Data & methodology: validity of approach, quality of data, quality of presentation

The authors have studied the effects of OA using a large variety of methods, ranging from molecular biophysics to in vivo behavioural tests. The data are of high quality, and the analysis seems adequate. The clarity of the presentation could be improved at some places (see H, below), but this should pose no problems to the authors. Please introduce page numbering into revised manuscript!

D. Appropriate use of statistics and treatment of uncertainties

For some of the reported significance values it is not clear what is compared to what (e.g., Fig. 1d, asterisk above blue bar). In general, for bar charts with several columns, it might be helpful to put the asterisk on top of a bracket that links a pair of bars, rather than on top of an individual bar. Are the fractional currents in OA significantly different for Ck (Fig. 4d, blue bar) and Ck-A558T (Fig. 5B, pink bar) channels, as indicated in the text?

E. Conclusions: robustness, validity, reliability

The conclusions seem to be well supported by the data. Based on the presented data the authors might even be able to make a statement on the kinetics of OA binding/unbinding to TRPV1. The slow time course ($\tau \sim 20-30$ s) over which the inhibitory effect of OA develops (Fig. 1e) suggests that the compound rates for the two-step process $OA(aq) \leftrightarrow OA(\text{membrane}) \leftrightarrow OA(\text{channel-bound})$ is slow. Because the first step of this process, partitioning of OA into the membrane, is likely the same regardless of the channel state, the difference in compound on-rates for closed and open channels (if significant) suggests that the second step (binding of membrane-dissolved OA to

the channel protein) might be rate limiting. Further, because the apparent inhibition rate is the sum of the compound on and off rates, the observed slower rate of inhibition in the open state suggests that the sum of the rates for binding and unbinding ($k_{on}+k_{off}$) of membrane-dissolved OA to the protein might be slower when the channel is open. On the other hand, OA stabilizes the closed state, therefore its K_d (k_{off}/k_{on}) must be smaller in the closed state. From these two arguments one might conclude that it is the on-rate of OA which is slowed in open channels. This might be too far of a shot, but I wonder whether the structure or the docking experiments might suggest anything along these lines...

F. Suggested improvements: experiments, data for possible revision

In Fig. 1d (red bar) ~55% current remains in 3.5 μ M OA, whereas in the dose response curve (Fig. 1f) only 10% current remains in 3 μ M OA. Weren't these experiments done under identical conditions (+120 mV, 4 μ M capsaicin)? If yes, what is the reason for this discrepancy?

Fig. 1e nicely shows the wash-on time course for OA. It might be informative to show the washoff time course as well.

In supplemental Table I it would be more user-friendly to plot the components in the order of increasing time constants. In the open-time fit for the after-OA condition (last row) two time constants are identical: this distribution should be fitted with only two components.

In Figure 2d the colors for the control and LPA conditions are hard to discern from each other.

The time constant of inhibition in Fig. 3f is 6 s, as opposed to 30 s in Fig. 1e. Might this difference be explained by the different voltages used (60 vs. 120 mV)?

G. References: appropriate credit to previous work?

No concerns.

H. Clarity and context: lucidity of abstract/summary, appropriateness of abstract, introduction and conclusions

In general, the logic of the paper is clear and easy to follow. Some small suggestions for improvement are as follows:

Abstract:

"whose pruritic activity through the activation of this ion channel can be inhibited by OA as well as that of histamine" -- change to -- "whose pruritic activity, as well as that of histamine, through the activation of this ion channel can be inhibited by OA"

Results:

"Since OA stabilizes a closed state we hypothesized that it could also bind to some site on the TRPV1 channel (as opposed to producing this effect by altering membrane properties)" -- I don't understand the meaning of "since" here: there is no logical connection between the two clauses.

Methods:

"The time-course of inhibition of TRPV1 currents by OA was obtained by applying a continuous voltage pulse of +60 mV in the presence of capsaicin and of capsaicin + OA."

This sentence contrasts with the following sentence: "Time courses of inhibition by 5 μ M OA were obtained at 120 mV."

and with the legend for Fig. 1e which also states +120 mV.

Fig. 4 legend: "The percentage of interaction of TRPV1 protein with OA was obtained by

densitometric analysis of the spots and normalized to the positive control or LPA."

Maybe instead:

"The percentage of interaction of TRPV1 protein with OA or LPA was obtained by densitometric analysis of the spots and normalized to the positive control."

Or did I misunderstand something here?

Reviewer #2 (Remarks to the Author):

The authors (Morales-Lázaro et al.) describe the property of oleic acid (OA) to inhibit TRPV1. This fatty acid, together with the structurally related petroselinic acid, is shown to inhibit in an allosteric manner the activation of TRPV1 by capsaicin, protons (pH 5.5) and cyclic phosphatidic acid (CPA). The binding site of OA is demonstrated to be in the capsaicin-binding pocket of TRPV1 (rat). This is investigated using the capsaicin-insensitive chicken TRPV1 ortholog and many TRPV1 point mutants. In vivo, OA diminished pain in mice receiving injections with capsaicin and itch in animals injected with CPA or histamine. The experiments are well designed and the manuscript well written. Nevertheless, there are a few important points that the authors should clarify and respectively improve.

1) The work of Matta et al. (2007) should be cited and a commentary given regarding the previous report, showing that OA minimally activates TRPV1, instead of inhibiting it, like reported here. Was there such activation (possibly transient) observed immediately after applying OA alone? All current recordings seem to be performed a certain time after the OA application, which is not presented. The authors should show a continuous current recording comprising the time just before, during and immediately after the OA application in the bath. The recording should be done preferably with repeated voltage ramps or at constant potentials (e.g. -80 and +80 mV) and also show the current elicited by capsaicin in the same cell.

In the same paper, OA (50 μ M) is described to have a very weak inhibitory effect over TRPV1's response to capsaicin (30 nM) and a very small effect over the current activated by pH 5.5. This is in large contrast to the dramatic inhibition reported here at concentrations 10 times lower. What could be the source of this difference between the two reports?

The authors should also mention the property of some polyunsaturated fatty acids, like DHA, of activating TRPV1 (the same paper) and of LNA and EPA of inhibiting TRPV1 currents elicited by capsaicin (somehow similarly to the reported effect of OA here).

Whether OA could inhibit TRPV1's activation by DHA or other n-3 PUFA would be interesting and relevant for a fatty acid mixture, but probably beyond the scope of this study.

2) One of the most important properties of TRPV1 is its thermal sensitivity.

Further experiments showing a possible modulation by OA of TRPV1's activation by temperature (>42 {degree sign}C) would add a lot to the paper's interest and possibly clarify if OA is also activating TRPV1 at higher, physiologically relevant temperatures, close to the threshold of TRPV1 activation.

In the eventuality that these experiments are not permitted by the authors' technical setup, at least the authors should discuss whether OA would modulate the activation of TRPV1 through this modality. This could be speculated based on the existing knowledge about how temperature activates TRPV1 and considering OA as an allosteric inhibitor of TRPV1. Perhaps, the work of Gavva et al. (2005), segregating TRPV1 synthetic antagonists in types A and B should be mentioned.

3) The authors report that both pain-related behavior (LPA- or capsaicin-evoked paw licking) and itch-related behavior (histamine-evoked hindlimb scratching of the neck) are inhibited by OA. A more direct comparison of antinociceptive and antipruritic effects of OA could have been tested using the "cheek" model which discriminates between pain and itch behaviors (Shimada & LaMotte, 2008).

References

Matta JA, Miyares RL, Ahern GP. J Physiol. TRPV1 is a novel target for omega-3 polyunsaturated fatty acids. 2007 578(Pt 2):397-411.

Gavva NR1, Tamir R, Klionsky L, Norman MH, Louis JC, Wild KD, Treanor JJ. Proton activation does not alter antagonist interaction with the capsaicin-binding pocket of TRPV1. Mol Pharmacol. 2005 68(6):1524-33.

Shimada SG, LaMotte RH. Behavioral differentiation between itch and pain in mouse. Pain. 2008;139(3):681-7.

Reviewer #3 (Remarks to the Author):

This manuscript reports the identification of oleic acid (OA) as a novel inhibitor of the temperature, pH- and capsaicin-sensing TRPV1 ion channel. This result is novel and interesting as OA is an endogenous lipid and therefore a potential regulator of TRPV1. The authors show by careful electrophysiological analysis and clever usage of TRPV1 orthologues and point mutations that OA inhibits TRPV1 by stabilizing its closed state, specifically by occupying the previously identified capsaicin binding site. Finally, they show that OA is inhibiting TRPV1 mediated pain and itch. Overall, this is a nice study, where all experiments are performed to a very high standard. The manuscript is well written and the figures well designed. I thus have only two minor points I would like to see addressed:

1) In order to put any potential physiological effect of OA into context with these results, the authors should discuss in detail what physiological and pathological concentrations of OA are. Would endogenous OA levels ever give rise to any relevant effects? And importantly, at what concentrations of OA are other ion channels and receptors affected in their function? In any case I find the results exciting as they are, but I would like to see a detailed section added to the discussion.

2) It is unclear to me how the voltage step data (Figure 3a) result in continuous GV curves presented in Figure 3b. I was expecting discrete points at each voltage step instead. Please, clarify this.

Reviewer #1

We thank this Reviewer for his/her comments which have improved our manuscript. Below are our responses to your comments.

“This group has largely contributed to the identification of natural TRPV1 activators in the past. Thus, in addition to vanilloids, acidic pH, heat, and voltage, they have identified several lipid activators. However, in contrast to the large repertoire of known natural activators, only few natural TRPV1 inhibitors are known. In addition to a better understanding of physiological processes, discovering TRPV1 inhibitors might have clinical relevance for the treatment of pain and itch. The discovery of OA as a TRPV1 inhibitor is therefore of considerable interest, and is likely to appeal to a broad readership. I recommend the paper for publication following minor revision.”

1) **“For some of the reported significance values it is not clear what is compared to what (e.g., Fig. 1d, asterisk above blue bar). In general, for bar charts with several columns, it might be helpful to put the asterisk on top of a bracket that links a pair of bars, rather than on top of an individual bar.”**

Thank you. We have done this and have also introduced page numbering into the manuscript.

2) **“Are the fractional currents in OA significantly different for Ck (Fig. 4d, blue bar) and Ck-A558T (Fig. 5B, pink bar) channels, as indicated in the text?”**

The data obtained (shown as uninhibited currents) for the chicken (Ck; 0.71 ± 0.13) and the Ck-A558T mutants (0.31 ± 0.11) are statistically different ($p = 0.0047$, t-test).

3) **“The conclusions seem to be well supported by the data. Based on the presented data the authors might even be able to make a statement on the kinetics of OA**

binding/unbinding to TRPV1. The slow time course ($\tau \sim 20-30$ s) over which the inhibitory effect of OA develops (Fig. 1e) suggests that the compound rates for the two-step process $OA(aq) \leftrightarrow OA(\text{membrane}) \leftrightarrow OA(\text{channel-bound})$ is slow. Because the first step of this process, partitioning of OA into the membrane, is likely the same regardless of the channel state, the difference in compound on-rates for closed and open channels (if significant) suggests that the second step (binding of membrane-dissolved OA to the channel protein) might be rate limiting. Further, because the apparent inhibition rate is the sum of the compound on and off rates, the observed slower rate of inhibition in the open state suggests that the sum of the rates for binding and unbinding ($k_{on} + k_{off}$) of membrane-dissolved OA to the protein might be slower when the channel is open. On the other hand, OA stabilizes the closed state, therefore its K_d (k_{off}/k_{on}) must be smaller in the closed state. From these two arguments one might conclude that it is the on-rate of OA which is slowed in open channels. This might be too far of a shot, but I wonder whether the structure or the docking experiments might suggest anything along these lines... ”

The analysis suggested by the reviewer is, in principle, correct, although we note that the equilibrium between partitioning of OA into the membrane and the binding to the channel might be more complex than a two state process. We have carried out new docking simulations of OA onto the open and closed states of TRPV1. The reported binding energy of OA to the closed state is -5.25 kcal/mol and to the open state is -4.91 kcal/mol, suggesting that binding to the closed state is indeed more favorable. We note however that these are equilibrium binding energies and in order to get an idea of the energy barriers involved in binding and unbinding, a molecular dynamics simulation of the binding-unbinding process is the appropriate procedure, but this is beyond the scope of this work.

Moreover, we have added a sentence at the end of page 7 where we state that: “As the estimated charge (q_b) remains relatively constant around a value of 0.5 (Fig. 3c and d), we can calculate that the closed state is favored by about 1 kcal/mol”.

4) “In Fig. 1d (red bar) ~55% current remains in 3.5 μ M OA, whereas in the dose response curve (Fig. 1f) only 10% current remains in 3 μ M OA. Weren't these experiments done under identical conditions (+120 mV, 4 μ M capsaicin)? If yes, what is the reason for this discrepancy?”

We thank you for pointing this out. The experiment we had previously shown on Fig. 1d was **not** done under identical conditions to the one shown in Fig.1f. There was a mistake in the figure legend and 3.5 μ M, a sub-saturating concentration of OA, was applied only for 3 mins rather than 5 mins. For this reason we have replaced this figure with another using a continuous 40 mV pulse that shows that the intracellular application of 5 μ M OA rapidly inhibits TRPV1 currents (Fig. 1a). Also shown is the recovery from inhibition. By performing the experiment in this manner we can resolve the time course for reversibility from inhibition (as suggested by this reviewer in comment 5 below).

5) “Fig. 1e nicely shows the wash-on time course for OA. It might be informative to show the wash-off time course as well.”

We thank the Reviewer for suggesting this experiment which we now have performed. The data are shown in Figs. 1a and b.

6) “In supplemental Table I it would be more user-friendly to plot the components in the order of increasing time constants. In the open-time fit for the after-OA condition (last row) two time constants are identical: this distribution should be fitted with only two components.”

Again, thank you for the suggestion. We have now reordered the contents of Table 1 and fitted the open times in OA to just two components.

7) “In Figure 2d the colors for the control and LPA conditions are hard to discern from each other.”

Thank you. We have changed the colors for the traces to make them easier to visualize.

8) “The time constant of inhibition in Fig. 3f is 6 s, as opposed to 30 s in Fig. 1e. Might this difference be explained by the different voltages used (60 vs. 120 mV)?”

This is a good point. We have now substituted the time courses of inhibition previously shown in Figure 1 for time courses obtained at 60 mV applying OA in the presence of capsaicin (as is the case for the single channel recordings) and at -60 mV applying OA in the absence of capsaicin. We had previously shown inhibition in the open and closed states at 120 mV but we believe it is more accurate to show inhibition at a depolarizing and at a hyperpolarizing voltage in the presence and absence of an agonist. In the open state, at 60 mV, inhibition by OA proceeds with a time course similar to that obtained with single channels. (7.7 s) while in the closed state, inhibition is faster (2.9 sec) as we had previously observed. We have now eliminated former Supplemental Figure 1 in which we showed inhibition at 120 mV after 5 min of OA application since the new experiments provide a clearer insight of the state-dependence of inhibition by OA.

The Reviewer has asked us to clarify the following sentences:

9) "whose pruritic activity through the activation of this ion channel can be inhibited by OA as well as that of histamine" -- change to -- "whose pruritic activity, as well as that of histamine, through the activation of this ion channel can be inhibited by OA"

We have made the following change: “Moreover, we report a novel itch-inducing molecule, cyclic phosphatidic acid (cPA), that activates TRPV1 and whose pruritic activity, as well as that of histamine, occurs through the activation of this OA inhibitable ion channel.”

10) **“Since OA stabilizes a closed state we hypothesized that it could also bind to some site on the TRPV1 channel (as opposed to producing this effect by altering membrane properties)” -- I don't understand the meaning of "since" here: there is no logical connection between the two clauses.”**

We have rephrased this sentence and changed it to the beginning of the “Interactions of OA with TRPV1” section on page 8. It reads as follows: “Most of the other structurally similar compounds that were tested (Supplementary Fig. 3) did not exhibit the inhibitory properties of OA on TRPV1 activation. For this reason, we hypothesized that OA would bind to a specific site on TRPV1 (as opposed to producing its effect by altering membrane properties)^{17,18} .

11) **“The time-course of inhibition of TRPV1 currents by OA was obtained by applying a continuous voltage pulse of +60 mV in the presence of capsaicin and of capsaicin + OA.”**

This sentence contrasts with the following sentence: "Time courses of inhibition by 5 μ M OA were obtained at 120 mV and with the legend for Fig. 1e which also states +120 mV.”

As discussed in points 5 and 8, we have now performed recovery from inhibition experiments at 40 mV and time courses of inhibition for the open state at 60 mV and for the closed state at -60 mV. We have now been clarified this in the Methods section (please see pages 22-23).

12) **“Fig. 4 legend: "The percentage of interaction of TRPV1 protein with OA was obtained by densitometric analysis of the spots and normalized to the positive control or LPA.”**

Maybe instead: "The percentage of interaction of TRPV1 protein with OA or LPA was obtained by densitometric analysis of the spots and normalized to the positive control." Or did I misunderstand something here?”

The reviewer did not misunderstand and we thank him/her for this suggestion. We have now changed the sentence to read “The percentage of interaction of TRPV1 protein with OA or LPA was obtained by densitometric analysis of the spots and normalized to the positive control which was LPA itself.” Please see page 38, Figure legend 4f.

Reviewer #2

We thank the Reviewer for careful assessment of our manuscript.

“The authors (Morales-Lázaro et al.) describe the property of oleic acid (OA) to inhibit TRPV1. This fatty acid, together with the structurally related petroselinic acid, is shown to inhibit in an allosteric manner the activation of TRPV1 by capsaicin, protons (pH 5.5) and cyclic phosphatidic acid (CPA). The binding site of OA is demonstrated to be in the capsaicin-binding pocket of TRPV1 (rat). This is investigated using the capsaicin-insensitive chicken TRPV1 ortholog and many TRPV1 point mutants. In vivo, OA diminished pain in mice receiving injections with capsaicin and itch in animals injected with CPA or histamine. The experiments are well designed and the manuscript well written. Nevertheless, there are a few important points that the authors should clarify and respectively improve.”

1) “The work of Matta et al. (2007) should be cited and a commentary given regarding the previous report, showing that OA minimally activates TRPV1, instead of inhibiting it, like reported here. Was there such activation (possibly transient) observed immediately after applying OA alone? All current recordings seem to be performed a certain time after the OA application, which is not presented. The authors should show a continuous current recording comprising the time just before, during and immediately after the OA application in the bath. The recording should be done preferably with repeated voltage ramps or at constant potentials (e.g. -80 and +80 mV) and also show the current elicited by capsaicin in the same cell.”

The trace in Figure 4B of the Matta et al. (2007) study show that there is a small increase in current-magnitude when 50 μM oleic acid and pH 5.5 are co-applied. On page 402 of the Matta et al., article it is stated, “In contrast, mono-unsaturated oleic acid (50 μM) failed to potentiate proton-evoked responses, reflecting the poor agonistic properties of this lipid at TRPV1 (Fig. 4B and C).” We note however, that the error bars in Fig. 4C show that, on average, there is a decrease in the currents as compared to those evoked at pH 5.5. In our study, we show that under our experimental conditions, 5 μM OA does **not** produce activation of TRPV1 currents (please see below for description of the experiment). Nonetheless, we have now mentioned and cited this article in our manuscript (please see pages 18-19).

As suggested by the reviewer, we now show that that 5 μM OA does **not** activate TRPV1 (see Supplementary Fig. 1). The traces in this figure show that using continuous pulses of +40 mV and/or -40 mV and by adding 5 μM OA right after washing off capsaicin, that OA does not transiently activate TRPV1 but rather, inhibits the currents through this channel when capsaicin is reapplied.

2) “In the same paper, OA (50 μM) is described to have a very weak inhibitory effect over TRPV1's response to capsaicin (30 nM) and a very small effect over the current activated by pH 5.5. This is in large contrast to the dramatic inhibition reported here at concentrations 10 times lower. What could be the source of this difference between the two reports?”

From what is stated in the Methods section of the Matta et al. (2007) article, the preparation of oleic acid was different from ours. We sonicated our stock OA and vortexed recording solutions that contained OA. This might represent an important difference as we have noticed with the preparation of OA and lysophosphatidic acid that sonication is hugely important to ensure reproducible results pertaining either inhibition or activation of the TRPV1 channel. The work by Matta et al. (2007) was performed using oocytes and voltage-clamp rather than excised patch clamp, which is another difference with our experiments.

3) “The authors should also mention the property of some polyunsaturated fatty acids, like DHA, of activating TRPV1 (the same paper) and of LNA and EPA of inhibiting TRPV1 currents elicited by capsaicin (somehow similarly to the reported effect of OA here). Whether OA could inhibit TRPV1's activation by DHA or other n-3 PUFA would be interesting and relevant for a fatty acid mixture, but probably beyond the scope of this study. Beyond the scope”

Indeed, it would be interesting to study whether OA can inhibit the activation of TRPV1 by DHA or other omega-3 polyunsaturated fatty acids and we agree with the reviewer that this is beyond the scope of our study.

4) “One of the most important properties of TRPV1 is its thermal sensitivity. Further experiments showing a possible modulation by OA of TRPV1's activation by temperature (>42 °C) would add a lot to the paper's interest and possibly clarify if OA is also activating TRPV1 at higher, physiologically relevant temperatures, close to the threshold of TRPV1 activation.

In the eventuality that these experiments are not permitted by the authors' technical setup, at least the authors should discuss whether OA would modulate the activation of TRPV1 through this modality. This could be speculated based on the existing knowledge about how temperature activates TRPV1 and considering OA as an allosteric inhibitor of TRPV1.”

We thank the reviewer for suggesting this experiment. In this regard, using a method we previously reported to study TRPV1 activation by temperature (Islas et al. J. Neurosci. Methods. 2015, 243: 120-5) we have performed this experiment by applying fast temperature ramps. Figure 2h-j shows that 5 μ M OA markedly inhibits ($72 \pm 8\%$) activation of TRPV1 by temperature.

5) “Perhaps, the work of Gavva et al. (2005), segregating TRPV1 synthetic antagonists in types A and B should be mentioned.”

We have added a section to the Discussion section where we review the results of Gavva et al. (2005) in light of our own findings. Please see page 18 where we state: “Gavva and collaborators had previously proposed a classification of TRPV1 antagonists based on their ability to inhibit activation of this channels by capsaicin and protons (group A) or on their ability to inhibit activation by capsaicin but not protons (group B)⁴¹. These authors stated that group A antagonists lock the TRPV1 channels in the closed state; which is consistent with what we find with OA and TRPV1”.

6) “The authors report that both pain-related behavior (LPA- or capsaicin-evoked paw licking) and itch-related behavior (histamine-evoked hindlimb scratching of the neck) are inhibited by OA. A more direct comparison of antinociceptive and antipruritic effects of OA could have been tested using the "cheek" model which discriminates between pain and itch behaviors (Shimada & LaMotte, 2008).”

We have performed these experiments and the results are shown in Supplementary Figure 7. Our data indicate that capsaicin injection produces an increase in pain-related behavior (wiping bouts) and no increase in itch-related behavior (scratching bouts), as compared to animals injected only with vehicle solutions (Supplementary Figure 7a and b). With histamine injections we found that this compound does not produce pain-related behavior (Supplementary Figure 7c) but it increases itch-related behavior significantly, as compared to the control (Supplementary Figure 7d). For OA, we found that injection of this fatty acid alone, produced no significant pain- or itch-related responses (Supplementary Figures 7 a-d). However, when OA was injected together with capsaicin, a clear decrease in pain-related behavior was observed (Supplemental Figure 7a) and, when OA was injected together with histamine, it partially obliterated the effects of histamine on itch-related behavior (Supplementary Figure 7d). These new data are in agreement with our previous results shown in Figure 6 and Supplementary Figure 6.

We have added all of the references shown below.

Matta JA, Miyares RL, Ahern GP. J Physiol. TRPV1 is a novel target for omega-3 polyunsaturated fatty acids. 2007 578(Pt 2):397-411.

Gavva NR1, Tamir R, Klionsky L, Norman MH, Louis JC, Wild KD, Treanor JJ. Proton activation does not alter antagonist interaction with the capsaicin-binding pocket of TRPV1. Mol Pharmacol. 2005 68(6):1524-33.

Shimada SG, LaMotte RH. Behavioral differentiation between itch and pain in mouse. Pain. 2008;139(3):681-7.

Reviewer #3

We thank this Reviewer for valuable comments and suggestions.

“This manuscript reports the identification of oleic acid (OA) as a novel inhibitor of the temperature, pH- and capsaicin-sensing TRPV1 ion channel. This result is novel and interesting as OA is an endogenous lipid and therefore a potential regulator of TRPV1. The authors show by careful electrophysiological analysis and clever usage of TRPV1 orthologues and point mutations that OA inhibits TRPV1 by stabilizing its closed state, specifically by occupying the previously identified capsaicin binding site. Finally, they show that OA is inhibiting TRPV1 mediated pain and itch. Overall, this is a nice study, where all experiments are performed to a very high standard. The manuscript is well written and the figures well designed. I thus have only two minor points I would like to see addressed”

1) “In order to put any potential physiological effect of OA into context with these results, the authors should discuss in detail what physiological and pathological concentrations of OA are. Would endogenous OA levels ever give rise to any relevant effects? And importantly, at what concentrations of OA are other ion channels and

receptors affected in their function? In any case I find the results exciting as they are, but I would like to see a detailed section added to the discussion.”

To address this concern, we have added the following paragraphs to the Discussion section of our manuscript:

“In mammalian astrocytes albumin stimulates OA synthesis that promotes neuronal differentiation and serves as a neurotrophic factor⁴⁴. Both albumin and OA can be incorporated into neurons and lead to dendritic growth through the activation of the PPAR α receptor⁴⁵. Moreover, the albumin-OA complex has been actually proposed for the treatment of paralysis, spasticity and pain⁴⁶. With regard to OA’s concentrations under physiological conditions it was found that in human blood physiological concentrations for OA range between 10-100 μM ⁴⁷ and its concentration increases under ischemic conditions⁴⁸.

There are only few reports that describe the effects of OA on the biophysical properties of ion channels. While K_{ACh} channels are inhibited by 5 μM OA and BK_{Ca} channels are activated by 10 μM ²⁸, $\text{Kv}7.2/3$ channels remain unaffected by 70 μM OA²⁹. For sodium and calcium channels in CA1 neurons³⁰ and for potassium channels in taste cells³¹ 16 μM and 10 μM , respectively, have little or no effect. For human cardiac Nav1 channels and gap junction channels in smooth muscle cells, 5 μM and 25 μM OA, respectively, display inhibitory effects^{32,33}. In human skeletal muscle sodium channel currents (hSkM1) are increased by 5 μM OA³⁴. Finally, in trigeminal neurons 30 μM OA increases Ca_i^{2+} influx⁴⁸”.

We have added this information to the Discussion section (please see page 19).

Finally, we have also added a paragraph discussing a recent report by Gao et al., 2016 in which a new TRPV1 channel structure was obtained showing the presence of a phospholipid in the vanilloid binding pocket (please see page 17).

2) “It is unclear to me how the voltage step data (Figure 3a) result in continuous GV curves presented in Figure 3b. I was expecting discrete points at each voltage step instead. Please, clarify this”.

We thank the Reviewer for pointing this out. We now show the graph in Figure 3b with its corresponding discrete points rather than with continuous lines.

REVIEWERS' COMMENTS:

Reviewer #1 (Remarks to the Author):

The authors have constructively addressed all my concerns. I find the present form of the manuscript ready for publication.

Reviewer #2 (Remarks to the Author):

The manuscript and supplementary material are much-improved. There are only a few minor comments/corrections.

In the new version of Figure 1 and in the Supplementary Figure 1, capsaicin is applied more than once. The authors should state or show how repeated capsaicin challenges on inside-out patches under Ca²⁺-free conditions affect the current reproducibility. Most likely tachyphylaxis shouldn't be a concern, but commenting on or showing this will help increase the confidence in the presented results.

Minor comments

page 18 (solution preparations): "isometric" instead of "isotonic"; I presume they wanted to say "isotonic" conditions, isometric has no meaning in the context or is seldom used with this meaning in English (it refers to solutions, not to muscle contractions)

"capsaicin and OA co-applied", I presume "were" is missing (bottom, page 22)

"TRPV1proteins" without space (page 26)

"Image J" instead of "ImageJ" (page 26)

Reviewer #3 (Remarks to the Author):

My few comments have been well addressed. This is now a very nice manuscript.

Reviewer #2

We thank this Reviewer for his/her careful review of our manuscript and comments which have improved our manuscript. Below are our responses to your comments.

“The manuscript and supplementary material are much-improved. There are only a few minor comments/corrections.

In the new version of Figure 1 and in the Supplementary Figure 1, capsaicin is applied more than once. The authors should state or show how repeated capsaicin challenges on inside-out patches under Ca²⁺-free conditions affect the current reproducibility. Most likely tachyphylaxis shouldn't be a concern, but commenting on or showing this will help increase the confidence in the presented results.”

We have now added two insets to Supplementary Figure 1 where we show that repeated applications of capsaicin do not produce desensitization/tachyphylaxis of the TRPV1 channel under our experimental conditions and at the two voltages shown (+40 and -40 mV). We have also explained this result on page 5 of the manuscript.

Furthermore, we have fixed all of the typos and replaced the word isometric (page 22) by the word isotonic as suggested by the reviewer.